# Mitochondrial Function, Mobility and Lifespan Are Improved in *Drosophila melanogaster* by Extracts of 9-*cis*-β-Carotene from *Dunaliella salina*

**DOI:** 10.3390/md17050279

**Published:** 2019-05-10

**Authors:** Tobias Weinrich, Yanan Xu, Chiziezi Wosu, Patricia J. Harvey, Glen Jeffery

**Affiliations:** 1Institute of Ophthalmology, University College London, 11-43 Bath Street, London EC1V 9EL, UK; t.weinrich@ucl.ac.uk; 2School of Science, University of Greenwich, Central Avenue, Chatham ME4 4TB, UK; Y.Xu@greenwich.ac.uk (Y.X.); Chiziezi.Wosu@greenwich.ac.uk (C.W.); P.J.Harvey@greenwich.ac.uk (P.J.H.)

**Keywords:** ageing, 9-*cis*-β-carotene, mitochondrial function, mobility, lifespan, *Drosophila melanogaster*, microalgae, *Dunaliella salina*

## Abstract

Carotenoids are implicated in alleviating ageing and age-related diseases in humans. While data from different carotenoids are mixed in their outcomes, those for 9-*cis*-β-carotene indicate general positive effects, although basic data on its biological impact are limited. Here, we show that supplementation with 9-*cis*-β-carotene in ageing *Drosophila melanogaster* improved mitochondrial function in terms of ATP production and whole-body respiration and extended mean lifespan. It also resulted in improved mobility. These data provide a potential biological rational for the beneficial effects of dietary supplementation with 9-*cis-*β-carotene. These effects may be based on the maintenance of a sound mitochondrial function.

## 1. Introduction

Ageing is regulated by intrinsic and extrinsic factors [1]. Harman [2] proposed that a key driver of the ageing process is declining mitochondrial function that results in reduced production of ATP, the energy source that underpins cellular function. This in turn is associated with increased production of reactive oxygen species (ROS) and formation of a hyperoxidant state that contributes to systemic inflammation and increases the pace of ageing [1,2].

Diet is a key extrinsic factor that impacts on both the quality of life and its length. Carotenoids are natural organic pigments found in all photosynthetic organisms, including algae, and are able to interact with free radical and oxygen singlets. They are not synthesized in animals who depend on dietary intake to maintain an adequate supply of carotenoids. Maintaining carotenoid levels has many biological advantages, as they are strong anti-oxidants because of their conjugated double-bond structure and have anti-tumour and anti-inflammatory properties [3]. Despite evidence of their positive impact in the prevention of age-related diseases, results of human trials remain controversial. In part, this is due to their numerous types and diverse nature. Further, their use in human trials suffers from confounding issues associated with human heterogeneity. Consequently, it is important to assess their impact in tightly controlled experiments that reduce these variables.

Carotenoids are characterised by a polyene backbone consisting of a series of conjugated C=C bonds: alterations in the backbone modifying the number of conjugated double bonds or the addition of chemical functional groups alter their reactivity [4]. Carotenes such as α- and β-carotene contain exclusively carbon and hydrogen atoms, whilst xanthophylls such as lutein and zeaxanthin also contain oxygen. Multiple geometric (*cis*/*trans* or *Z*/*E*) isomers are also possible, although the more stable all-*trans* configuration is more common, and methods for the synthetic production of all-*trans* but not *cis* isomers are well established [5,6].

*Dunaliella salina* (*D. salina)* is one of the richest sources of natural carotenoids and accumulates a high content of β-carotene (up to 10% of the dry biomass), of which 9-*cis*-β-carotene makes up ~50% of total β-carotene. Recent studies with algal preparations enriched in 9-*cis*-β-carotene have proved to be promising, showing beneficial effects on retinal dystrophies such as retinitis pigmentosa [7] as well as atherosclerosis, diabetes, and psoriasis [8,9,10]; also, Sher et al. [11] demonstrated positive effects using synthetic 9-*cis*-β-carotene. However, we do not know what the impact of 9-*cis*-β-carotene has on fundamental mitochondrial metrics, which are known to be involved in the regulation of ageing.

Hence, here we investigated the effect of *D. salina* extracts rich in 9-*cis*-β-carotene on lifespan, mobility and mitochondrial function of *Drosophila melanogaster*. We also asked which components of the extract were responsible for any improvements in these metrics.

## 2. Results

HPLC profiles of the different carotenoid extracts are shown in Figure 1a. The identity of 9-*cis*-β-carotene was verified by UV-Vis spectral features (maxima in nm at 424, 446 and 472 nm) coupled to the retention time relative to all-*trans* β-carotene determined by HPLC (Figure 1b,c) and by mass spectrometric analysis (*m*/*z* = 536.439). All-*trans*-β-carotene and 9-*cis*-β-carotene are the most abundant carotenes in algal extracts, with only small amounts of other carotenoids, including lutein, zeaxanthin, α-carotene, and phytoene. Their concentrations are shown in Table 1. The two algal extracts obtained after supercritical CO_2_ extraction had similar profiles but differ slightly in the relative amounts of the two β-carotene isomers. The supercritical CO_2_ extracts had a 9-*cis*-/all-*trans*-β-carotene ratio of ~0.8 (sample 1) and ~1.4 (sample 2) and reflected seasonal batch-to-batch variation in carotenoid productivity of the alga grown outdoors. By contrast the high 9-*cis*-β-carotene extract had a very high 9-*cis*-/all*-trans*-β-carotene ratio of ~2.4 and a much lower relative concentration of α-carotene. As extracts were painted onto the surfaces of corn-yeast agar meal, the concentration of carotenoids in corn-yeast agar meal was also analysed. No carotenes were detected, and the concentrations of lutein (0.007 µg mL^−1^) and zeaxanthin (0.003 µg mL^−1^) were negligible compared with the amounts present in the extracts.

The impact of *D. salina* carotenoid extract on lifespan was investigated and compared with those of synthetic all-*trans*-β-carotene and un-extracted *D. salina* powder along with controls. This is shown in Figure 2a,b. In males (Figure 2a), there was an overall increase in the median lifespan when they were fed the extracts. However, these differences were only significant for 10 µM total β-carotene solutions, which induced a 32.6% increase in median life span. No concentration determined an increase in absolute lifespan, as animals in both experimental and control groups were dead by around day 80. Hence, the positive impact was on increasing the probability of survival in middle-aged and late-middle-aged flies. However, treatment with un-extracted *D. salina* powder had a significant detrimental effect on lifespan, decreasing the median lifespan by approximately 25% compared with controls. In female flies (Figure 2b), there was also a significant increase in median lifespan with the 10 µM carotenoid solution, although of a slightly lower magnitude than in male flies. However, unlike in male flies, the un-extracted *D. salina* powder had no impact on females.

Treatments that improve lifespan are often associated with functional improvements. When seven-week-old, the same groups as above that were examined for lifespan were tested for mobility by assessing negative geotaxis following one week of supplementation. Figure 2c,d shows the locomotor function at 8 weeks of age for the different treatment groups and for both sexes. As with lifespan, the impact of the extract was greater in males than in females, but in both sexes, the solution applied at 10 µM with respect to total β-carotene was the most effective dose at improving the climbing ability, although, in males, this was matched by the 100 µM solution. Again, as with lifespan, the un-extracted *D. salina* had a significant negative impact on males but not on females when compared with the controls. A similar observation of negative effects using whole algal biomass was reported by Ross and Dominy [12], who showed that whilst biomass supplements prepared from washed and dried *Spirulina platensis* had a deleterious effect on the growth of chicks fed 10 and 20% algae as part of their diet, *Spirulina biomass* up to 12% could substitute for other protein sources in broiler diets and would mediate good growth and feed efficiency.

These combined results identified algal extracts with 10 µM total carotene as the most effective concentration for improved lifespan and mobility. Hence, we treated male flies, in which the effects were more marked, to assess the impact of natural carotenoid supplementation upon mitochondria (Figure 3). This time, we used a *D. salina* extract (DS Extract 2) which contained a greater concentration of 9-*cis*-β-carotene relative to all-*trans*-β-carotene and set the concentration of all-*trans*-β-carotene to match that used in the previous experiment. Whole-body metabolic rate was measured via CO_2_ production, and total body ATP levels were also measured. Whole metabolic rate is used as a surrogate marker for mitochondrial respiration. In treated flies, the metabolic rate increased by 27% (Figure 3a), and ATP levels increased by 26.5% compared with the control group (Figure 3b). Consequently, these data are consistent with natural β-carotene extracts from *D. salina* enriched in 9-*cis*-β-carotene improving mitochondrial function.

*D. salina* extracts contain different carotenoids, including all-*trans*-β-carotene, 9-*cis*-β-carotene and lutein. To confirm the identity of the component responsible for the observed biological improvements, we fed flies with the different carotenoids and assessed locomotor function (Figure 4). Because of the inability to get a stable purified 9-*cis*-β-carotene sample, an extract with very high 9-*cis*-/all-*trans*-β*-*carotene ratio was used. The results confirmed that 9-*cis*-β-carotene was responsible for the biological improvements.

## 3. Discussion

This study demonstrates that 9-*cis*-β-carotene derived from extracts of a natural alga improves mitochondrial function, mean lifespan and mobility in *D. melanogaster*. Nagpal and Abraham [13] recently showed that a synthetic sample of all-*trans*-β-carotene exerted antigenotoxic and antioxidant effects in *Drosophila*; however 9-*cis*-β-carotene has a higher antioxidant activity than all-*trans*-β-carotene and may also be more efficient than all-*trans*-β-carotene in vivo [14,15].

The potential protective effects of different carotenoids have been the subject of debate, with data both in favour of their use and against it. However, they have not analysed the combined metrics used in this study. The amplitude of the effects on lifespan was different in females and males, indicative of physiological differences between sexes. This sex difference was highlighted when flies were treated with only the un-extracted *D. salina* powder. This resulted in a significant reduction in lifespan and locomotor activity in males but had less impact on females.

The carotenoid literature is complex, with a large number of variables between studies. Studies vary in the application of different carotenoids at different concentrations on different species and over different periods. Treatments have been tested in models in which the physiology of the organism has been challenged in some way via external agents and in natural models where there has been no external challenge. Further, while lifespan has often been examined, other readouts have varied. This makes a coherent synthesis of the literature difficult to perform. Experiments on lifespan in mice failed to find significant differences [16], while experiments in invertebrates appeared more promising, including those in *Caenorhabditis elegans* [17] and *Tenebrio molitor* [18]. Positive results have been found in *Drosophila* in some studies [19,20] but not others [21]. However, these studies have not examined 9-*cis*-β-carotene.

It is thought that carotenoids play a role in the protection of the human central retina because they produce pigments that protect against short-wavelength light. β-carotene is strongly associated with this process. However, the overall evaluation of carotenoids in humans has been clouded by the association of synthetic all-*trans*-β-carotene with lung cancer [22]. This has significantly marred the development of carotenes for human health. However, our study has revealed consistent biological improvement in flies using the geometric isomer 9-*cis*-β-carotene derived from the marine microalga *D. salina*, and this may have wider cross-species translation. There is a growing literature across species showing that 9-*cis*-β-carotene supplementation has a positive impact. In mice, it prevents atherosclerotic progression when the animals are given a high-fat diet [23], and this is consistent with the demonstration that, in humans, it increases high-density lipoprotein (HDL)-cholesterol [24]. In humans suffering from retinitis pigmentosa, a degenerative disease of the retina, where disrupted mitochondrial dysfunction is implicated [25,26], 9-*cis*-β-carotene improves retinal function [7]. Likewise, in fundus albipunctatus, which is a retinal dystrophy, 9-*cis*-β-carotene again improved retinal function [27], whilst Sher et al. [11] showed that 9-*cis*-β-carotene inhibited photoreceptor degeneration in cultures of eye cup receptors. That these are retinal studies may be significant, because the retina has the greatest concentration of mitochondria in the body and shows a marked decline of mitochondria with age [28], leaving significant room for therapeutic action to any agent that has the ability to improve mitochondrial function. 

In spite of the highly complex nature of the carotene literature, there is now clear evidence that potentially positive impacts may derive from the use of 9-*cis*-β-carotene, from fly through mouse and to man. These are likely due to its influence over mitochondrial function. Given the pivotal role of mitochondria in ageing and many diseases, either as a primary or a secondary mechanism, this may be of significance.

## 4. Materials and Methods

### 4.1. Carotenoid Extracts

These were derived from biomass of *D. salina* cultured in open raceway ponds by Monzon Biotech, Spain, harvested, then freeze-dried and extracted using supercritical CO_2_ by NATECO_2_ (Germany). The extracts were diluted in ethanol before use. DS extract 1 and DS extract 2 were extracted from two batches of biomass that were harvested in different seasons, and consequently they present minor differences in the relative composition of all-*trans-* and 9-*cis*-β-carotene, as shown in Table 1. Further carotenoid extracts containing very high 9-*cis*-β-carotene content were prepared in laboratory incubators according to a patented cultivation process with red light treatment (GB2017019440-Production of *Dunaliella*) of *D. salina* strain CCAP 19/41. The biomass was harvested in batches in 50 mL centrifugation tubes by centrifugation at 3000× *g* for 10 min. The cells were ruptured via ultrasonication and vortexed in ethanol to provide an extract that was then clarified by centrifugation at 3000× *g* for 10 min, and the supernatant was aliquoted into 14 amber glass bottles for fly feeding. Lyophilised *D. salina* biomass which was not extracted was also used as a control. Lutein was from Holland and Barrett (Lutigold UK), and all-*trans*-β-carotene from Sigma-Aldrich (UK). 

The carotenoid composition of the extracts was analysed using high-performance liquid chromatography with diode array detection (HPLC–DAD). Each extract in ethanol was filtered through a 0.45 µm syringe filter into amber HPLC vials and then analysed using a YMC30 250 × 4.9 mm I.D S-5 µm HPLC column. The column temperature was set at 25 °C, and the flow rate of the mobile phase at 1 mL min^−1^, with isocratic elution with 80% methanol/20% methyl *tert*-butyl ether (MTBE) with a pressure of 88 bar. 9-*cis*-β-carotene was identified from the analysis of UV-Vis spectral features and from *m*/*z* data obtained using a Waters Acquity UPCC (Waters, Elstree, UK) instrument fitted with a Diode Array Detector and connected to a Synapt G2 HDMS (Waters, Elstree, UK). The Synapt G2 was fitted with an electrospray source and operated in positive ion mode over a mass range of 50–800 *m*/*z* units. Wavelength-dependent absorption was measured using the DAD, operating in the wavelength range 200–700 nm. Inlet conditions A: scCO_2_; B: methanol + 0.1% formic acid (*v*/*v*); make-up solvent/methanol + 0.1% formic acid (*v*/*v*); column: acquity UPLC HSS C18 SB, 3.0 × 100 mm, 1.8 µm particle size. Processing was carried out using MassLynx v4.1.

Absorbance at 450 nm was used to quantify all-*trans*-β-carotene, 9-*cis*-β-carotene, α-carotene and lutein, and absorbance at 280 nm was used to quantify phytoene. Carotenoids standards of synthetic β-carotene, α-carotene, lutein and phytoene were obtained from Sigma-Aldrich (UK), and the concentrations of 9-*cis*- and all-*trans*-β-carotene, α-carotene, lutein and phytoene in the extracts were estimated using the standard curves.

In initial experiments, four different concentrations of *D. salina* extract (DS Extract 1, ratio of 9-*cis*-/all-*trans*-β-carotene, 0.8) were dissolved in 100% ethanol to give a range of concentrations with respect to total β-carotene (9-*cis*- + all-*trans*-β-carotene), i.e., 100 μM, 10 μM, 1 μM and 0.1 μM. A further set of experiments was conducted with a *D. salina* extract (DS Extract 2, ratio 9-*cis*-/all-*trans*-β-carotene, 1.4), and with a highly concentrated 9-*cis* extract prepared from laboratory cultivation, (ratio 9-*cis*-/all-*trans-*β-carotene, 2.4), each used at the same relative concentration of all-*trans*-β-carotene in ethanol as for DS Extract 1. Synthetic all-*trans*-β-carotene was used at the same relative concentration of all-*trans*-β-carotene in ethanol as for DS Extract 1, and Lutigold was tested at the same concentration of lutein determined in DS Extract 1 (see Table 1). Finally, normal corn meal/agar/sucrose/yeast was substituted by a lyophilised *D. salina* compound in 1% agar, on a kcal like-for-like substitution. Volumes of 100 µL of each concentration were sprayed on top of normal food, for each group. The control group consisted of 100% ethanol only. The samples were feed every 2 days until the end of each experiment. 

### 4.2. Drosophila Melanogaster

Wild-type male and female *D. melanogaster* Dahomey were used. These were separately housed in standard fly vials containing a normal cornmeal/sugar/yeast/agar medium maintained at 25 °C, 12 h L/D. To determine the effects of the *D. salina* extract on longevity, 150 flies were collected for each experimental group (experimental groups *N* = 7, 30 flies per vial, with 5 vials per group) separated into males and females. The number of dead flies was counted three times per week, and the remaining flies were transferred to fresh vials. Survival patterns were calculated using the Kaplan–Meier method and are presented as survival curves. The mean, median, minimum and maximum lifespans were determined. 

Locomotor activity was assessed at eight weeks of age separately in males and females in the same groups as above (*N* = 7). The measurements were performed using the negative geotaxis assay following one week of diet supplementation. Briefly, the flies were placed in empty polystyrene vials marked at 8 cm of height and gently tapped to the bottom. The number of flies that climbed above the mark in 20 s after being tapped to the bottom were counted. The test was performed 10 times for each group of 10 flies. 

Whole-animal metabolic rate was assessed by measuring CO_2_ production in lab-made respirometers following a protocol previously described [29]. In each group, there were five replicates containing five flies. Metabolic rate was measured over 120 min. 

### 4.3. Data Analysis

Data collected were analysed with GraphPad Prism v.6, and statistical analysis was undertaken using two-way Mann–Whitney *U* test unless otherwise stated. A *p* < 0.05 value was considered significant, and data presented are mean + SEM.

## Figures and Tables

**Figure 1 marinedrugs-17-00279-f001:**
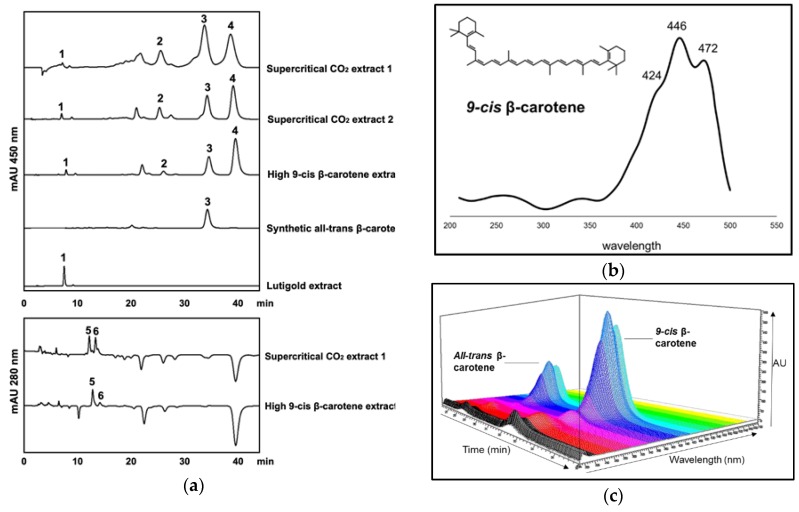
(**a**) HPLC chromatograms of ethanol extracts of carotenoids at 450 nm and 280 nm showing the carotenoid profiles. The major peaks shown are: 1, lutein; 2, α-carotene; 3, all-*trans*-β-carotene; 4, 9-*cis*-β-carotene; 5, 15-*cis*-phytoene; 6, all-*trans*-phytoene; mAU, milli-Absorbance Units. (**b**) UV-Visible spectrum of 9-*cis*-β-carotene; (**c**) 3D image showing spectral features and relative retention times for all-*trans*-β-carotene and 9-*cis*-β-carotene after HPLC chromatography.

**Figure 2 marinedrugs-17-00279-f002:**
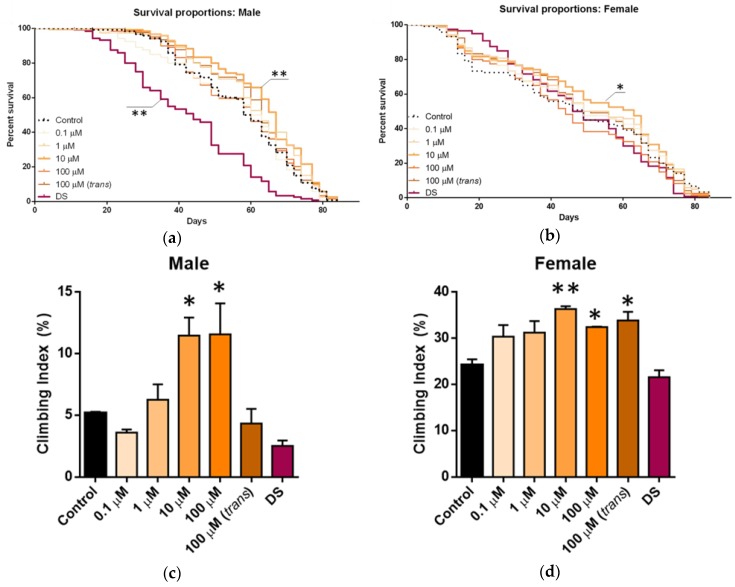
Lifespan (**a** and **b**) and locomotor activity (**c** and **d**) in flies treated with *D. salina* extract, *D. salina* powder and synthetic all-*trans*-β-carotene. DS Extract 1 was applied at different concentrations with respect to β-carotene content (varying from 0.1 to 100 µM) and synthetic all-*trans*-β-carotene (100 µM *trans*), and effect on lifespan of male flies (**a**) and females (**b**) was recorded (*n* = 120 flies in each group), Long-rank Mantel Cox test. In (**c**) and (**d**), locomotor activity assessed by negative geotaxis in seven-week-old flies treated since birth (*n* = 40 flies in each group), one-way ANOVA, multiple comparisons Bonferroni; * *p* < 0.05, ** *p* < 0.01).

**Figure 3 marinedrugs-17-00279-f003:**
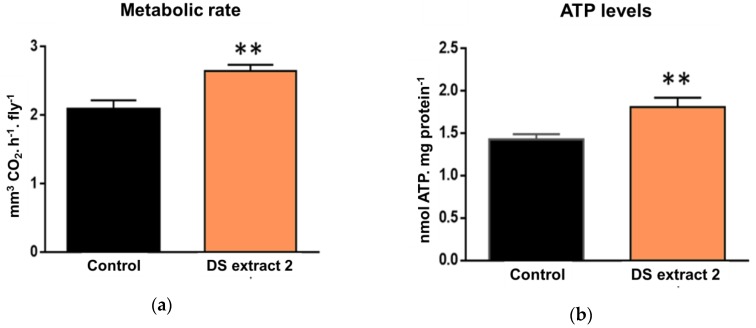
Effects of 14-day DS extract 2 supplementation in aged flies: metabolic rate (**a**) and ATP levels (**b**). The treatment with DS carotenoids extract enriched in *9-cis*-β-carotene significantly improved mitochondrial function compared with untreated flies; metabolic rate was measured as a marker of (**a**) mitochondrial respiration and (**b**) ATP levels. Six replicates containing five flies each. Results are mean + SEM; ** *p* < 0.01.

**Figure 4 marinedrugs-17-00279-f004:**
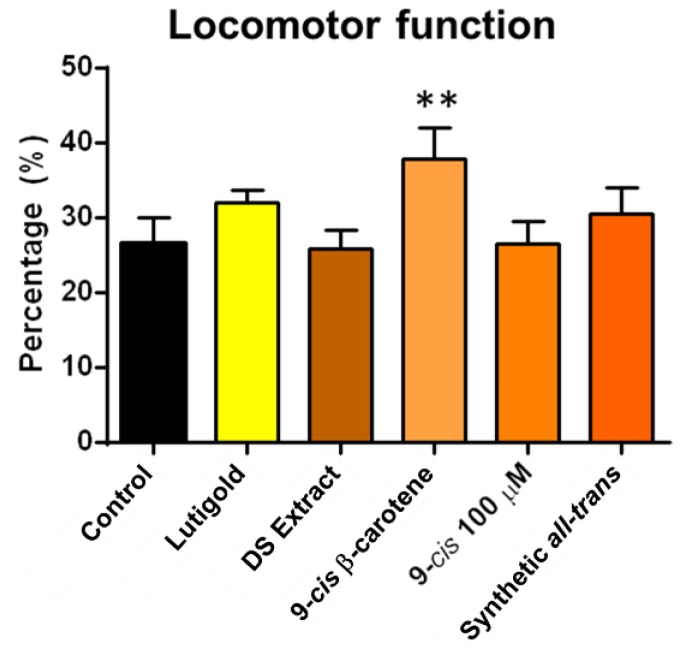
Fourteen-day supplementation in aged flies with different carotenoids from *D. salina,* DS Extract (6.71 µM all-*trans*-β-carotene and 8.94 µM 9-*cis*-β-carotene), 9-*cis*-β-carotene (5.59 µM all-*trans*-β-carotene and 13.22 µM 9-*cis*-β-carotene); Lutigold (0.07 µM lutein) and synthetic all-*trans*-β-carotene (5.59 µM all-*trans*-β-carotene). Effects on locomotor function assessed by negative geotaxis after administration of different carotenoids supplements. Only flies in the group treated with high concentration of 9-*cis*-β-carotene significantly improved their climbing index by 36.8%. All other groups had climbing indexes similar to that of the control group; *n* = 60 flies per group. Results are mean +SEM. ** *p* < 0.01.

**Table 1 marinedrugs-17-00279-t001:** Concentrations of individual carotenoids in the working ethanol extracts. Volumes of 100 µL of working ethanol extracts were sprayed onto the surface of fly feed every 2 days. DS, *Dunaliella salina*.

	Concentration of Carotenoid			
Test Sample	All-*trans*-β-carotene (µM)	9-*cis*-β-carotene (µM)	α-carotene (µM)	Lutein (µM)
Synthetic all-*trans*-β-carotene	5.59	-	-	-
Lutigold	-	-	-	0.07
Supercritical CO_2_ DS extract 1 (~10 µM with respect to carotene)	5.40	4.28	1.26	0.07
Supercritical CO_2_ DS extract 2	6.71	8.94	2.01	0.35
High 9-*cis*-β-carotene extract	5.59	13.22	0.88	0.44

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
