# Peer review of "Mitochondrial Function, Mobility and Lifespan Are Improved in Drosophila melanogaster by Extracts of 9-cis-β-Carotene from Dunaliella salina"

_marinedrugs, 2019, doi:10.3390/md17050279_

Reviewer 1 Report

In this study the authors described the potential biological role for aged improvements following dietary supplementation with 9-cis β-carotene based on modulation of mitochondrial metrics.

This manuscript was well organized in study design and well written, and which includes useful information for researchers about 9-cis β-carotene.

Their hypothesis and current findings are interesting. The experimental protocols were well controlled and data supports authors'observations.

Author Response

We thank the reviewer for their supportive comments

Reviewer 2 Report

The manuscript is centered on the evaluation of 9-cis β-carotene benefic effects in ageing Drosophila melanogaster.

1- Line 81: Explain what is the signification of the following sentence: At no concentration was there an increase in absolute lifespan

2-Line 125: Figure 4. 14 day supplementation in aged flies with different carotenoids from D. salina, commercial  Lutigold (Lutein) and synthetic all-trans β-carotene.

Please, give the concentration of all used compounds.

3-    Line 171: Carotenoid extracts. These were derived from biomass of D. salina cultured in open raceway. 

Line 173: Further carotenoid extracts containing very high 9-cis β- carotene content were prepared in laboratory incubators 

Please clarify the process of extraction and the reference of each extract (ED1 and ED2), in the Table 1, concentrations are given for CO2 DS extract 1 and CO2 DS extract 2, what is the reference of the laboratory extract? What is the difference between CO2 DS extract 1 and CO2 DS extract 2?

4- In general, graphs of results didn’t show the positive controls of ageing Drosophila melanogaster improvement

5- Discussion of results need to be improved, there are not reference to actual literature about the administration of carotene in drosophila.

Author Response

please see responses listed numerically below

We have extended the text to explain  the meaning of  absolute lifespan

At no concentration was there an increase in absolute lifespan as animals in all groups in both experimental and control groups were dead by around day 80. Hence, the positive impact is on increasing probability of survival in middle aged and late middle aged flies.

Concentrations are now given in the Figure Legend (4) as requested.

We have have provided extra information regarding the extraction, as requested.

The graphs show group separation and where significant the its degree via conventional symbols. In both Figure 2a and b the 10uM curve is well to the right of the others for the majority of its course.

We have included additional references to the Discussion as requested.     

Please note that changes to the text are marked in the re-submission

Reviewer 3 Report

nice paper

I am suggesting a few textual changes that may give some improvement to this already fine article.  I leave the suggestions to your discretion.

Lines 15 , 16, 17

These data provide a potential biological rational for beneficial effects of dietary supplementation of 9-cis beta-carotene.  These effects may be based on maintenance of sound mitochondrial function.

Line 30.  adequate supply of carotenoids.

Line 50.  which are known to be involved in regulation of ageing.

Line 67.  and the concentrations of the carotenoids lutein .....

Line 83.  say something in the Discussion about the negative effect mentioned here. It would contribute positively to the discussions on controversies regarding different results regarding effects of carotenes.

Lines 93 and 94 .  please say whether the " negative impact " is significant or not.

Line 145.  regarding " that have been challenged " :  Please say a bit more about this.  I don't know what is meant here.

Line 214  the numbers "140 flies" , N=7, and 30 flies per vial " do not seem to match up.  Please explain better.

Author Response

Line 30, corrected as requested

Line 50 (original), corrected as requested

Line 67 (original), corrected as requested

Line 83.........

Line 93 and 94 (original), we have stated significant, as requested.

Line 145 (original) we have expanded the text to signify the meaning of challenge

Line (original) 214. There was a typographic error which we are grateful for the reviewer revealing. It should have been 150 per group.

Please note that changes to the text in the re-submission are marked

Reviewer 4 Report

The manuscript "Mitochondrial function, mobility and lifespan are improved in Drosophila melanogaster by extracts of 9-cis β-carotene from Dunaliella salina" by Jeffery and coworkers describes effects of 9-cis β-carotene in ageing Drosophila melanogaster for the improvement of mitochondrial function in terms of ATP production that extends  mean lifespan. The method described in this manuscript is novel. The relevant references are mostly covered. The experiments are well designed and the Results are discussed with appropriate analytical evidence. In addition, I think this manuscript will be useful for broad range of readers. Therefore, I recommend this manuscript to publish in Marine Drugs.

Author Response

We thank the reviewer for their supportive comments

Round  2

Reviewer 2 Report

Accepted in the present form